# Decapsulation of Dextran by Destruction of Polyelectrolyte Microcapsule Nanoscale Shell by *Bacillus subtilis* Bacteria

**DOI:** 10.3390/nano10010012

**Published:** 2019-12-18

**Authors:** Egor V. Musin, Aleksandr L. Kim, Alexey V. Dubrovskii, Ekaterina B. Kudryashova, Sergey A. Tikhonenko

**Affiliations:** 1Institute of Theoretical and Experimental Biophysics Russian Academy of Science, Institutskaya st., 3, 142290 Puschino, Moscow Region, Russia; eglork@gmail.com (E.V.M.); kimerzent@gmail.com (A.L.K.); dav198@mail.ru (A.V.D.); 2G. K. Skryabin Institute of Biochemistry and Physiology of Microorganisms, Federal Research Center “Pushchino Scientific Center for Biological Research of the Russian Academy of Sciences”, Russian Academy of Sciences (Ibpm Ras), Prospekt Nauki, 5, 142290 Pushchino, Moscow Region, Russia; katryn@ibpm.pushchino.ru

**Keywords:** decapsulation, microcapsules, polyelectrolyte microcapsules, spore, *Bacillus subtilis*

## Abstract

One of the prerequisites of successful address delivery is controlling the release of encapsulated drugs. The new method of bacterial spore encapsulation in polyelectrolyte microcapsules allows for degrading the nanoscale membrane shell of microcapsules. The possibility of encapsulating spore forms of *Bacillus subtilis* in polystyrenesulfonate sodium/ polyallylamine hydrochloride (PSS/PAH) polyelectrolyte microcapsules was demonstrated. The activation and growth on a nutrient medium of encapsulated bacterial spores led to 60% degradation of the microcapsules nanoscale membrane shell. As a result, 18.5% of Fluorescein isothiocyanatedextran was encapsulated into polyelectrolyte microcapsules, and 28.6% of the encapsulated concentration of FITC-dextran was released into the solution.

## 1. Introduction

The main problem in the use of drugs in medicine is the lack of selectivity. This can lead to many side effects, as well as a decrease in the effectiveness of drugs. One of the ways to solve the problem is to encapsulate medicinal substances in polyelectrolyte microcapsules (PMC). PMCs can be obtained by layer-by-layer adsorption of oppositely charged polyelectrolytes on the surface of microparticles and have a diameter of several hundred nanometers to several microns [1]. The PMC shell allows protection of the encapsulated substance from the effects of aggressive environmental factors. For instance, encapsulation of urease allows preservation of the enzyme activity in a solution with proteinase K, while the non-encapsulated enzyme under these conditions is rapidly inactivated [2]. B. Sukhorukov et al. demonstrated preservation of the activity of encapsulated urease in the presence of proteinase K in solution, while the free enzyme rapidly lost activity under these conditions [3]. Due to such protection, PMC can be used for drug delivery both to the focus of the disease [4] and to individual cells of the body [5], with the ability to control the speed of their release [4].

The release of the encapsulated substance is usually achieved by destroying the integrity of the microcapsule. For example, Borodina T.N. et al. [6] showed the destruction of capsules formed from biodegradable polyelectrolytes under the action of proteolytic enzymes. Authors of various articles have shown the controlled destruction of microcapsules with a change in the pH value of the medium [7], as well as the additional structural elements included in them [8,9]. For example, Demina P.A. et al [10] showed that microcapsules, in the shell of which TiO_2_ nanoparticles are included, are destroyed by ultraviolet radiation. The research papers [8,11,12] showed the destruction of PMCs containing silver and gold or rhodamine nanoparticles by laser radiation, and the presence of Fe_3_O_4_ particles in the capsule shell makes it possible to control them by microwave radiation [9]. The described methods involve the use of installations that generate the corresponding radiation. It leads to complication of the process of destruction and its rise in price, so we propose a fundamentally new approach that allows you to destroy the shell PMC.

In this paper, we propose a fundamentally new approach to the destruction of the PMC shell through the germination of bacterial spores by aerobic endospore-forming bacterium *Bacillus subtilis* subsp. subtilis included in the microcapsule. Currently, researchers are actively working on the use of encapsulated forms of *B. subtilis*. For example, in [13], bacterial cells of *Bacillus subtilis* were encapsulated in alginate microcapsules, which improved the germination of cotton seeds. Shantanu S. Balkundi showed the possibility of encapsulating *B. subtilis* spores while maintaining their viability in polyelectrolyte layers of polyglutamic acid and polylysine, which were applied directly to the surface of the spores [14].

It is proposed to use the spores of *B. subtilis* not only as an object of encapsulation but also for breaking the nanoscale shell of a polyelectrolyte microcapsule. This method will allow the further release of medicinal substances contained in the cavity of PMC, with the destruction of the nanoscale shell by germinating bacterial spores when injected into the nutrient medium. 

## 2. Materials and Methods 

**Materials**. Polyelectrolytes polystyrenesulfonate sodium (PSS) and polyallylamine hydrochloride (PAA) with a molecular mass of 70 kDa, fluorescein isothiocyanate–dextran (150 kDa) Sigma (St. Louis, MS, USA), sodium chloride, ammonium sulfate, sodium carbonate, and calcium chloride from “Reahim” (Russian Federation, Lenin Region, St. Petersburg) were used. 

**Bacterial strains and their cultivation.** A typical strain of the species *Bacillus subtilis* subsp. subtilis VKM B-501^T^ All-Russian Collection of Microorganisms from Institute of Biochemistry and Physiology of Microorganisms (IBPM) Pushino was used. The culture was grown aerobically at 28 °C on tryptone soya broth (TSB) in test tubes for 3 days after which the tubes were kept at room temperature until spore formation (at least 90%). To encapsulate, a suspension was prepared in distilled water with a density of 15 × 10^8^/mL (according to McFarland).

**Preparation of fluorescently labelled PAA.** To a solution of polyelectrolyte (10 mg/mL) in 50 mM borate buffer, pH 9.0 with stirring (300–400 rpm), Fluorescein isothiocyanate (FITC) was slowly added in the same buffer. The components were fused in a molar ratio of FITC:PAH (by amino groups) = 1:100. After 1.5–2 h of incubation, the resulting solution was dialyzed against water (10 L) overnight.

**Preparation of CaCO_3_ microspherolytes**. 

We prepared three types of microspherolytes:Microspherolytes: To a 0.33 M CaCl_2_ solution vigorously agitated on a magnetic stirrer, an equal volume of 0.33 M Na_2_CO_3_ solution was rapidly added.Microspherolytes with spores: To a 0.33 M CaCl_2_ solution, containing 1.5 × 10^8^ bacterial spores vigorously agitated on a magnetic stirrer, an equal volume of 0.33 M Na_2_CO_3_ solution was rapidly added. Microspherolytes with spores and dextrain-FITC: To a 0.33 M CaCl_2_ solution, containing 1.5 × 10^8^ bacterial spores vigorously agitated on a magnetic stirrer, an equal volume of 0.33 M Na_2_CO_3_ solution, containing 6 mg/mL of FITC-dextran, was rapidly added.

Stirring was continued for 30 s. The resulting suspension was maintained until complete precipitation of the formed particles. The process of "ripening" of microspherolytes was controlled with the help of a light microscope. Then, the supernatant was decanted and the precipitate was washed with water and used to prepare PMC. The microparticles were obtained with a narrow size distribution with an average diameter of 8 ± 1 μm. The number of formed microspherolytes was determined in the Goryaev chamber.

**Preparation of polyelectrolyte microcapsules.** The preparation of PMC was carried out under aseptic conditions using sterile solutions. Polyelectrolyte microcapsules were obtained by alternately adsorbing oppositely charged polyelectrolytes on a dispersed microparticle (microspherolytes) with its subsequent dissolution. The alternate adsorption of PSS and PAA on the surface of CaCO_3_ microspherolytes [15,16,17] was carried out in polyelectrolyte solutions with a concentration of 2 mg/mL, containing 0.5 M NaCl solution. Each step of adsorption was followed by washing three times with a solution of 0.5 M NaCl, which is necessary for the removal of non-adsorbed polymer molecules. Particles were separated from the supernatant by centrifugation at 500 g. After applying six layers, carbonate cores were dissolved in a 0.2 M solution of EDTA for 2 h. The obtained capsules were washed three times with bi-distilled water to remove the destruction products of the CaCO_3_ core. 

**Determination of the viability of encapsulated spores and destruction of capsules.** One-hundred and 200 μL of a dense suspension of encapsulated spores were sown in 5 mL of 5.5 nutrient medium of TSB in test tubes and seeding in 5 mL of sterile distilled water served as a control. The tubes were incubated at 28 °C for 3 days. The presence of sediment and clouding of the medium indicated the destruction of the capsules and the growth of the culture. Confirmation of the presence of culture growth was performed by fluorescence and light microscopy, using a phase-contrast device. The study of the samples “bright-field microscopy” was carried out in a Zeiss Axiovert 200M Cell Observer microscope. To obtain fluorescent micrographs, excitation with light with a wavelength in the region of 480–490 nm was used using a combination of light filters.

**Registration of the yield of FITC-dextran from polyelectrolyte microcapsules.** The yield of the FITC-dextran from microcapsules was studied by fluorescence spectroscopy. FITC-dextran was decapsulated by the spore of *Bacillus subtilis*, and the suspension was precipitated by centrifugation at 15,000 rpm for 1 min. Further, the fluorescence of the supernatant was measured. Fluorescence spectra were recorded on an Infinite 200 Tecan instrument in a thermostated cuvette with an optical path length of 1 cm when excited with light at a wavelength of 480 nm.

## 3. Results and Discussion

The bacterial spores that we used are ellipsoidal [18]. 

Spores are highly resistant to many agents (chemical, thermal, and radiation) for prolonged exposure [19,20]. Figure 1b shows a micrograph of a spore of the strain VKM B-501^T^ in an optical microscope. 

*Bacillus subtilis* bacteria are non-pathogenic and members of the gut microflora, and for that reason, its spores are suitable for decapsulation of polyelectrolyte microcapsules in a digestive system [21,22].

Polyelectrolyte microcapsules were obtained with spores of the strain VKM B-501^T^ and included: particles in CaCO_3_ in the process of their formation, followed by deposition of polyelectrolyte layers (Figure 2). 

The method of light (Figure 3a) and fluorescence microscopy (Figure 3b) was used to analyze the products of encapsulation of spores, immediately after they were placed in a nutrient medium TSB, which indicated the absence of encapsulated spores. The capsules were particles with a diameter of 8 ± 1 μm, regular round shape, with a clearly defined undeformed shell (Figure 3). The shell thickness was 37 ± 3 nm.

*B. subtilis* growth in a liquid nutrient medium by seeding with a suspension of PMC, with the spores of the strain VKM B-501T, in a liquid nutrient medium tryptone soya broth (TSB) followed by incubation is shown in Figure 4. After 4 h of cultivation, spores began to germinate in a liquid nutrient medium (Figure 4). The picture showed that the spore started to destruct the nanoscale shell of the microcapsule and left the shell.

After 24 h of cultivation, growth was observed in the form of uniform turbidity of the nutrient medium. The study of the culture fluid by optical microscopy showed the presence of the same types of vegetative cells (number 2 at Figure 5a): whole microcapsules, deformed, germinating spores, and partially destroyed (number 1 at Figure 5a). As can be seen from Figure 5b, capsules have an irregular shape and are deformed. At the same time, the study, after 24 h of incubation of the control sowings in sterile distilled water PMC, with bacterial spores included, showed a complete absence of bacterial growth. Optical microscopy established the preservation of PMC, with bacterial spores included, unchanged.

As can be seen in Figure, spores germinated (number 2 at Figure 6) and went out from polyelectrolyte microcapsules (number 1 in Figure 6). The shell of 60% of the microcapsules is destroyed. Also, it can be seen that some microcapsules are intact. It can be related to the size of the polyelectrolyte microcapsules. The size of whole microcapsules is around 2–3 microns, and the size of the destructed microcapsules is 5–7 microns. 

Thus, it is abundantly clear that when released into the nutrient medium, the spores encapsulated in the PMC are activated and germinate, thus destroying the microcapsule nanoscale shells. 

We encapsulated 2.22 mg FITC-dextran into polyelectrolyte microcapsules, and thus, 18.5% was encapsulated. FITC-dextran was decapsulated after the destruction of the microcapsule shell by the spores. Six-hundred-and-thirty micrograms of FITC-dextran were released into the solution (28.6% of the encapsulated concentration).

## 4. Conclusions

We have found that the spores of *B. subtilis* subsp. can be encapsulated in the cavity of polyelectrolyte microcapsules. At the same time, the spores maintain their viability and, when they enter the nutrient medium, germinate, since the nanoscale shell of the microcapsules is semipermeable and capable of passing nutrients into the capsules. In the process of germination of bacterial spores, encapsulated in the cavity of the microcapsules, they deform and break the nanoscale shell of 60% of microcapsules, leaving the surrounding solution. Overall, 28.6% of encapsulated FITC-dextran was released into the solution after shell destruction of polyelectrolyte microcapsules. Thus, it can be said that the encapsulation of bacterial spores can be used to decapsulate and release the drugs contained inside the microcapsules under given conditions. The main advantage of this method of opening microcapsules is that it does not require the use of specialized expensive equipment, such as ultrasonic or laser generators.

Given the widespread use in medicine and veterinary medicine of *B. subtilis*, as a probiotic culture with antibacterial and antifungal effects, the results can be a practical foundation for creating a new form of drug that will allow the delivery of not only the medicinal substance protected from the negative effects of an aggressive environment but also a bacterial culture that restores the resident microflora in certain parts of the digestive tract.

## Figures and Tables

**Figure 1 nanomaterials-10-00012-f001:**
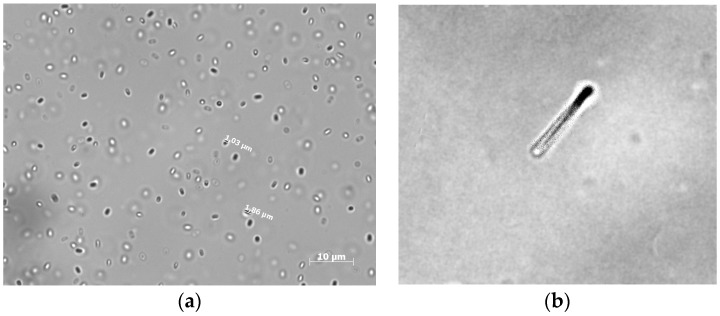
(**a**) Micrograph of *B. subtilis* spores in a light microscope. (**b**) Micrograph of germination of *B. subtilis* spores in a light microscope.

**Figure 2 nanomaterials-10-00012-f002:**
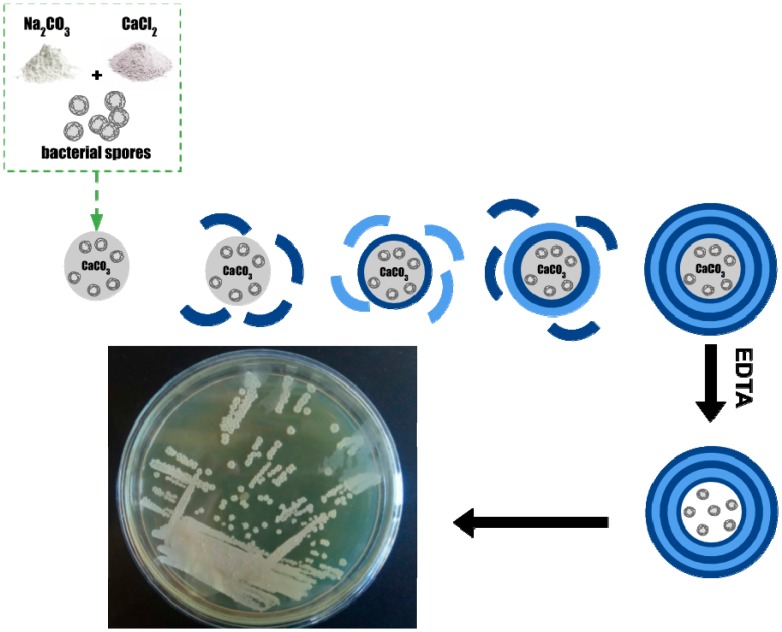
Diagram of the formation of polyelectrolyte microcapsules. Encapsulating spores of *B. subtilis* bacteria on CaCO_3_ particles.

**Figure 3 nanomaterials-10-00012-f003:**
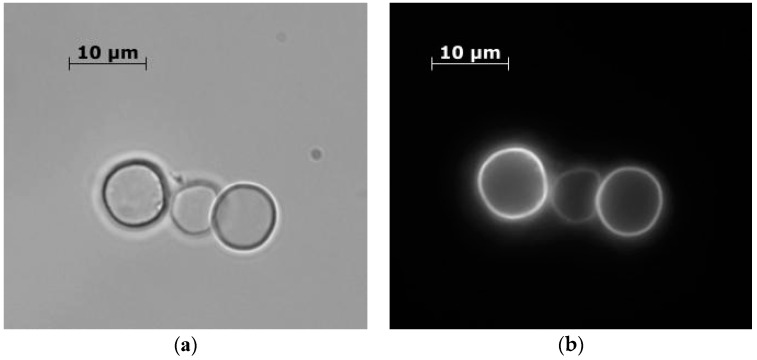
Polyelectrolyte microcapsules containing spores of *B. subtilis* subsp. subtilis, in a nutrient medium before incubation. (**a**) optical microscopy, (**b**) fluorescence microscopy.

**Figure 4 nanomaterials-10-00012-f004:**
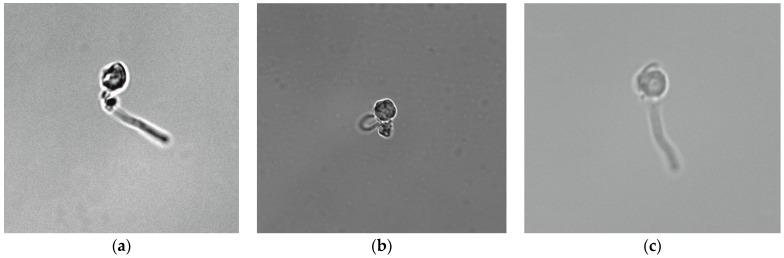
The destruction of polyelectrolyte microcapsule shell by spore germination (**a**–**c**).

**Figure 5 nanomaterials-10-00012-f005:**
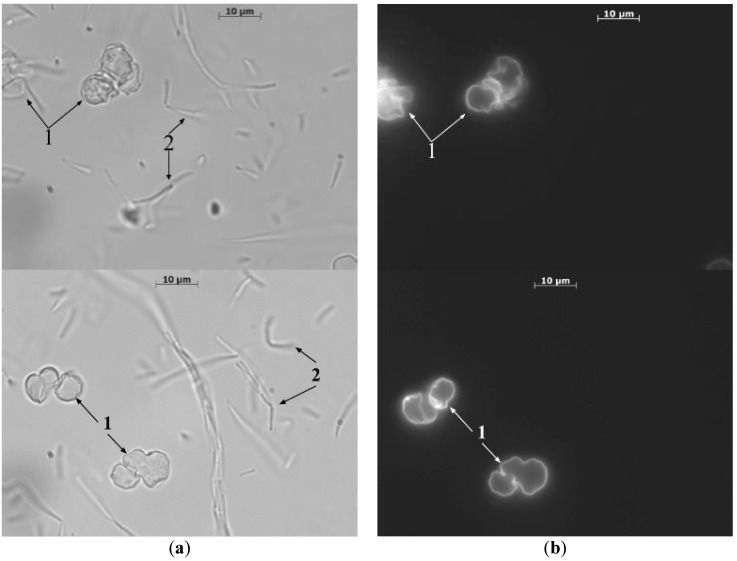
Photos of the sample culture fluid after 24 h of incubation of PMC in a nutrient medium. (**a**)—light microscopy; (**b**)—fluorescence microscopy. Optical (**a**) and fluorescent microscopic (**b**) images of the culture fluid after 24 h of incubation of PMC in a nutrient medium. 1—polyelectrolyte microcapsules; 2—germinated spores.

**Figure 6 nanomaterials-10-00012-f006:**
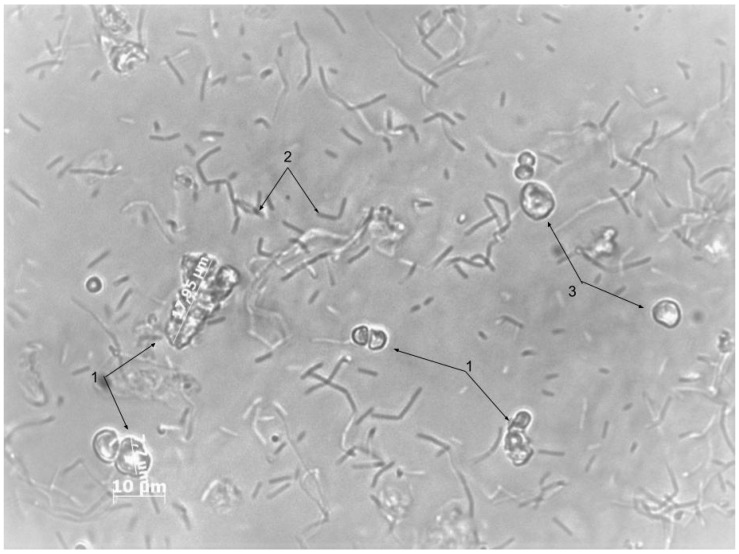
Photos of the sample culture fluid after 24 h of incubation of PMC in a nutrient medium. 1—Polyelectrolyte microcapsules with destructed shell; 2—vegetative cells of *B. subtilis*; 3—Polyelectrolyte microcapsules with the whole shell.

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
