# Peer review of "Decapsulation of Dextran by Destruction of Polyelectrolyte Microcapsule Nanoscale Shell by Bacillus subtilis Bacteria"

_nanomaterials, 2019, doi:10.3390/nano10010012_

Round 1

Reviewer 1 Report

The manuscript by S. Tikhonenko et al covers a fairly promising approach towards destruction of polyelectrolyte microcapsule shells by bacterial spores. Successful degradation of the microcapsules caused by encapsulated spores has been shown by microscopic studies (qualitatively) and via release of fluorescent labelled dextran as a model encapsulated compound (quantitatively). The introductory section provides sufficient background and includes relevant references. The experimental section contains important details and adequately describes used methods. The interpretation of the results is scientifically sound. Some minor stylistic and punctuation corrections may improve the quality of the presentation; they can be provided by the technical editor.

I support publication of the manuscript with minor text corrections.

Author Response

Dear Reviewer, thank you for taking the time to review our work. 

We made changes to the punctuation and English grammar of our article.

Reviewer 2 Report

I think that this new version of the manuscript 

can be accepted for publication in this journal.

Author Response

Dear Reviewer, thank you for taking the time to review our work.

Reviewer 3 Report

The manuscript is improved over the previous version. Some minor edits are indicated below.

1.  Lines 163-165:  Incomplete sentence. Suggest removing "A characteristic" from the beginning of the sentence and adding "is shown in Fig. 4" at the end of the sentence.

2.  Reference 2 is lacking journal name and pages.

3.  Reference 7 is lacking the volume number and page numbers.

4.  Reference 18 is lacking the page numbers.

5.  Reference 21 is missing part of the article title.

6.  Line 190:  Not sure what you mean by "solid". Intact or largely intact?

Author Response

Dear Reviewer, thank you for taking the time to review our work. 

We have made all the corrections.

1.  Lines 163-165:  Incomplete sentence. Suggest removing "A characteristic" from the beginning of the sentence and adding "is shown in Fig. 4" at the end of the sentence.

We have made changes in accordance with the recommendations.

2.  Reference 2 is lacking journal name and pages.

Reference 2 is patent. 

3.  Reference 7 is lacking the volume number and page numbers.

We added the volume number and page numbers

4.  Reference 18 is lacking the page numbers.

Reference 18 is a book.

5.  Reference 21 is missing part of the article title.

We added the article title.

6.  Line 190:  Not sure what you mean by "solid". Intact or largely intact?

We changed "solid" to "intact".

This manuscript is a resubmission of an earlier submission. The following is a list of the peer review reports and author responses from that submission.

Round 1

Reviewer 1 Report

The manuscript by Tikhonenko et al. reports on the attempts to decompose polyelectrolyte hollow microspheres (“microcapsules”) by the encapsulated spores of B. subtilis encapsulated. The topic of selective decomposition of polyelectrolyte hollow microspheres is actual from application viewpoint (drug delivery) and of fundamental interest. Unfortunately, the authors could not provide convincing proof of such decomposition. The microphotographs presented in Figures 3 and 4 show only several capsules. The presence of the spores encapsulated in them (Figure 3) is not documented. A circular object located between the left and central capsule cannot be identified as the spore based on this photo. Perhaps the encapsulation can be proven by selective fluorescent staining of the spores in the future. The degradation of hollow microspheres is shown in Figure 4. However, the evidence that the decomposition is caused by the spores is missing. The control experiments, both positive and negative, are required to prove the following statement: “… it can be said that the encapsulation of bacterial spores can be used to decapsulate and release the drugs contained inside the microcapsules under given conditions.”

Other drawbacks include: the title in not adequate; Illustrations are of poor quality (Figure 2); no copyright information presented for Figure 1; some statements are repeated.

I cannot recommend this manuscript for publication in the present state.

Reviewer 2 Report

This is an interesting manuscript. My comments are the following:

 1.- In general, Bacillus Subtilis bacteria are not pathogenic. However, digestive manifestations due to a high proliferation of bacteria, such as gastroenteritis, have been observed in isolate cases. Moreover, Bacillus Subtilis is related to the formation of a toxin (subtilina) that can produce allergic reactions in some patients. Why the authors decided to use this strain of bacteria?

2.-BSA was labeled with fluorescein isothiocyanate. However, BSA is a fluorescence protein by itself. Why has BSA been marked? The characteristics of the dialyzer used in the mark process have not been indicated in the text (line 70).

3.-Line 80: In my opinion, the preparation of polyelectrolyte microcapsules is not well indicated. The reader must know how the different layers of polymers are added to the capsules.

4.-Line 91: What is 5.5 nutrient medium of IBPM? Perhaps the pH is 5.5. This must be clarified.

5.-Figure 2 must be clarified. The first part of such a figure is not clear. I image that this fist part corresponds to the formation of CaCO3 microspherolytes.

These points should be clarified before the publication of the manuscript.           

Reviewer 3 Report

The manuscript by Musin and colleagues describes the degradation of polyelectrolyte microcapsules by Bacillus subtilis spores upon their germination in nutrient broth. There have been several studies involving microencapsulation of cells and spores of different species of Bacillus with consistent reports of viability of the encapsulated organisms and their ability to replicate in suitable growth media. Therefore, the study is not particularly novel. The justification of this approach is that spore germination and bacterial outgrowth is described as a simpler method to release encapsulated drugs than existing approaches. While this may very well be true, it was not demonstrated in a real-world drug delivery scenario, but rather by simply immersing the encapsulated spores in a bacterial growth medium. Proving that this approach would work in an animal model, would substantially strengthen the study and its conclusions.

There is no quantification provided in this study, nor indications of how many times the study was repeated. Were all of the microcapsules degraded? If not, what percentage of them?

Line 65:  Should be superscript 8

Line 96:  Not sure what you mean by: study of the drug “crushed drop”.

Lines 104-106:  The IC and OC labels in this figure refer to inner and outer spore coats, respectively. The term “shell” is incorrect here.

Line 108:  spores, rather than a spore.